# Backpropagation for Energy-Efficient Neuromorphic Computing

**Steve K. Esser**

IBM Research–Almaden

650 Harry Road, San Jose, CA 95120

sesser@us.ibm.com

**Rathinakumar Appuswamy**

IBM Research–Almaden

650 Harry Road, San Jose, CA 95120

rappusw@us.ibm.com

**Paul A. Merolla**

IBM Research–Almaden

650 Harry Road, San Jose, CA 95120

pameroll@us.ibm.com

**John V. Arthur**

IBM Research–Almaden

650 Harry Road, San Jose, CA 95120

arthurjo@us.ibm.com

**Dharmendra S. Modha**

IBM Research–Almaden

650 Harry Road, San Jose, CA 95120

dmodha@us.ibm.com

## Abstract

Solving real world problems with embedded neural networks requires both training algorithms that achieve high performance and compatible hardware that runs in real time while remaining energy efficient. For the former, deep learning using backpropagation has recently achieved a string of successes across many domains and datasets. For the latter, neuromorphic chips that run spiking neural networks have recently achieved unprecedented energy efficiency. To bring these two advances together, we must first resolve the incompatibility between backpropagation, which uses continuous-output neurons and synaptic weights, and neuromorphic designs, which employ spiking neurons and discrete synapses. Our approach is to treat spikes and discrete synapses as continuous probabilities, which allows training the network using standard backpropagation. The trained network naturally maps to neuromorphic hardware by sampling the probabilities to create one or more networks, which are merged using ensemble averaging. To demonstrate, we trained a sparsely connected network that runs on the TrueNorth chip using the MNIST dataset. With a high performance network (ensemble of 64), we achieve 99.42% accuracy at 108 $\mu$J per image, and with a high efficiency network (ensemble of 1) we achieve 92.7% accuracy at 0.268 $\mu$J per image.

## 1 Introduction

Neural networks today are achieving state-of-the-art performance in competitions across a range of fields [1][2][3]. Such success raises hope that we can now begin to move these networks out of the lab and into embedded systems that can tackle real world problems. This necessitates a shift

---

DARPA: Approved for Public Release, Distribution Unlimited

in thinking to system design, where both neural network and hardware substrate must collectively meet performance, power, space, and speed requirements.

On a neuron-for-neuron basis, the most efficient substrates for neural network operation today are dedicated neuromorphic designs [4][5][6][7]. To achieve high efficiency, neuromorphic architectures can use *spikes* to provide event based computation and communication that consumes energy only when necessary, can use *low precision synapses* to colocate memory with computation keeping data movement local and allowing for parallel distributed operation, and can use *constrained connectivity* to implement neuron fan-out efficiently thus dramatically reducing network traffic on-chip. However, such design choices introduce an apparent incompatibility with the backpropagation algorithm [8] used for training today's most successful deep networks, which uses continuous-output neurons and high-precision synapses, and typically operates with no limits on the number of inputs per neuron. How then can we build systems that take advantage of algorithmic insights from deep learning, and the operational efficiency of neuromorphic hardware?

As our main contribution here, we demonstrate a learning rule and a network topology that reconciles the apparent incompatibility between backpropagation and neuromorphic hardware. The essence of the learning rule is to train a network offline with hardware supported connectivity, as well as continuous valued input, neuron output, and synaptic weights, but values constrained to the range $[0, 1]$. We further impose that such constrained values represent probabilities, either of a spike occurring or of a particular synapse being on. Such a network can be trained using backpropagation, but also has a direct representation in the spiking, low synaptic precision deployment system, thereby bridging these two worlds. The network topology uses a progressive mixing approach, where each neuron has access to a limited set of inputs from the previous layer, but sources are chosen such that neurons in successive layers have access to progressively more network input.

Previous efforts have shown success with subsets of the elements we bring together here. Backpropagation has been used to train networks with spiking neurons but with high-precision weights [9][10][11][12], and the converse, networks with trinary synapses but with continuous output neurons [13]. Other probabilistic backpropagation approaches have been demonstrated for networks with binary neurons and binary or trinary synapses but full inter-layer connectivity [14][15].

The work presented here is novel in that i) we demonstrate for the first time an offline training methodology using backpropagation to create a network that employs spiking neurons, synapses requiring less bits of precision than even trinary weights, and constrained connectivity, ii) we achieve the best accuracy to date on MNIST (99.42%) when compared to networks that use spiking neurons, even with high precision synapses (99.12%) [12], as well as networks that use binary synapses and neurons (97.88%) [15], and iii) we demonstrate the network running in real-time on the TrueNorth chip [7], achieving by far the best published power efficiency for digit recognition (4 $\mu$J per classification at 95% accuracy running 1000 images per second) compared to other low power approaches (6 mJ per classification at 95% accuracy running 50 images per second) [16].

## 2 Deployment Hardware

We use the TrueNorth neurosynaptic chip [7] as our example deployment system, though the approach here could be generalized to other neuromorphic hardware [4][5][6]. The TrueNorth chip consists of 4096 cores, with each core containing 256 axons (inputs), a $256 \times 256$ synapse crossbar, and 256 spiking neurons. Information flows via spikes from a neuron to one axon between any two cores, and from the axon to potentially all neurons on the core, gated by binary synapses in the crossbar. Neurons can be considered to take on a variety of dynamics [17], including those described below. Each axon is assigned 1 of 4 *axon types*, which is used as an index into a lookup table of *s-values*, unique to each neuron, that provides a signed 9-bit integer synaptic strength to the corresponding synapse. This approach requires only 1 bit per synapse for the on/off state and an additional 0.15 bits per synapse for the lookup table scheme.

## 3 Network Training

In our approach, we employ two types of multilayer networks. The *deployment network* runs on a platform supporting spiking neurons, discrete synapses with low precision, and limited connectivity.

The *training network* is used to learn binary synaptic connectivity states and biases. This network shares the same topology as the deployment network, but represents input data, neuron outputs, and synaptic connections using continuous values constrained to the range $[0, 1]$ (an overview is provided in Figure 1 and Table 1). These values correspond to probabilities of a spike occurring or of a synapse being "on", providing a means of mapping the training network to the deployment network, while providing a continuous and differentiable space for backpropagation. Below, we describe the deployment network, our training methodology, and our procedure for mapping the training network to the deployment network.

## 3.1 Deployment network

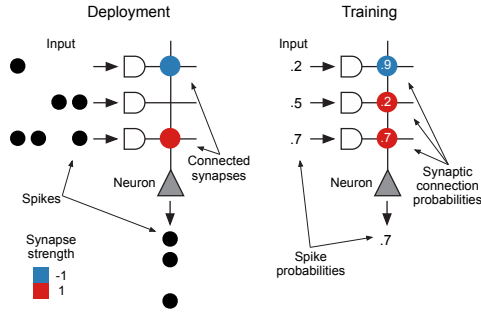

Figure 1: Diagram showing input, synapses, and output for one neuron in the deployment and training network. For simplicity, only three synapses are depicted.

Our deployment network follows a feed-forward methodology where neurons are sequentially updated from the first to the last layer. Input to the network is represented using stochastically generated spikes, where the value of each input unit is 0 or 1 with some probability. We write this as $P(x_i = 1) \equiv \tilde{x}_i$, where $x_i$ is the spike state of input unit $i$ and $\tilde{x}_i$ is a continuous value in the range $[0, 1]$ derived by re-scaling the input data (pixels). This scheme allows representation of data using binary spikes, while preserving data precision in the expectation.

Summed neuron input is computed as

$$I_j = \sum_i x_i c_{ij} s_{ij} + b_j, \tag{1}$$

where $j$ is the target neuron index, $c_{ij}$ is a binary indicator variable representing whether a synapse is on, $s_{ij}$ is the synaptic strength, and $b_j$ is the bias term. This is identical to common practice in neural networks, except that we have factored the synaptic weight into $c_{ij}$ and $s_{ij}$, such that we can focus our learning efforts on the former for reasons described below. The neuron activation function follows a history-free thresholding equation

$$n_j = \begin{cases} 1 & \text{if } I_j > 0, \\ 0 & \text{otherwise.} \end{cases}$$

These dynamics are implemented in TrueNorth by setting each neuron's leak equal to the learned bias term (dropping any fractional portion), its threshold to 0, its membrane potential floor to 0, and setting its synapse parameters using the scheme described below.

We represent each class label using multiple output neurons in the last layer of the network, which we found improves prediction performance. The network prediction for a class is simply the average of the output of all neurons assigned to that class.

Table 1: Network components

| | Deployment Network | | | Training Network | |
|---|---|---|---|---|---|
| | Variable | Values | Correspondance | Variable | Values |
| Network input | $x$ | $\{0, 1\}$ | $P(x = 1) \equiv \tilde{x}$ | $\tilde{x}$ | $[0, 1]$ |
| Synaptic connection | $c$ | $\{0, 1\}$ | $P(c = 1) \equiv \tilde{c}$ | $\tilde{c}$ | $[0, 1]$ |
| Synaptic strength | $s$ | $\{-1, 1\}$ | $s \equiv s$ | $s$ | $\{-1, 1\}$ |
| Neuron output | $n$ | $\{0, 1\}$ | $P(n = 1) \equiv \tilde{n}$ | $\tilde{n}$ | $[0, 1]$ |

## 3.2 Training network

Training follows the backpropagation methodology by iteratively i) running a forward pass from the first layer to the last layer, ii) comparing the network output to desired output using a loss function, iii) propagating the loss backwards through the network to determine the loss gradient at each synapse and bias term, and iv) using this gradient to update the network parameters. The training network forward pass is a probabilistic representation of the deployment network forward pass.

Synaptic connections are represented as probabilities using $\tilde{c}_{ij}$, where $P(c_{ij} = 1) \equiv \tilde{c}_{ij}$, while synaptic strength is represented using $s_{ij}$ as in the deployment network. It is assumed that $s_{ij}$ can be drawn from a limited set of values and we consider the additional constraint that it is set in "blocks" such that multiple synapses share the same value, as done in TrueNorth for efficiency. While it is conceivable to learn optimal values for $s_{ij}$ under such conditions, this requires stepwise changes between allowed values and optimization that is not local to each synapse. We take a simpler approach here, which is to learn biases and synapse connection probabilities, and to intelligently fix the synapse strengths using an approach described in the Network Initialization section.

Input to the training network is represented using $\tilde{x}_i$, which is the probability of an input spike occurring in the deployment network. For neurons, we note that Equation 1 is a summation of weighted Bernoulli variables plus a bias term. If we assume independence of these inputs and have sufficient numbers, then we can approximate the probability distribution of this summation as a Gaussian with mean

$$\mu_j = b_j + \sum_i \tilde{x}_i \tilde{c}_{ij} s_{ij}$$

and variance

$$\sigma_j^2 = \sum_i \tilde{x}_i \tilde{c}_{ij}(1 - \tilde{x}_i \tilde{c}_{ij})s_{ij}^2. \tag{2}$$

We can then derive the probability of such a neuron firing using the complementary cumulative distribution function of a Gaussian:

$$\tilde{n}_j = 1 - \frac{1}{2}\left[1 + \mathrm{erf}\left(\frac{\theta - \mu_j}{\sqrt{2\sigma_j^2}}\right)\right], \tag{3}$$

where erf is the error function, $\theta = 0$ and $P(n_j = 1) \equiv \tilde{n}_j$. For layers after the first, $\tilde{x}_i$ is replaced by the input from the previous layer, $\tilde{n}_i$, which represents the probability that a neuron produces a spike.

A variety of loss functions are suitable for our approach, but we found that training converged the fastest when using log loss,

$$E = -\sum_k \left[y_k \log(p_k) + (1 - y_k)\log(1 - p_k)\right],$$

where for each class $k$, $y_k$ is a binary class label that is 1 if the class is present and 0 otherwise, and $p_k$ is the probability that the the average spike count for the class is greater than $0.5$. Conveniently, we can use the Gaussian approximation in Equation 3 for this, with $\theta = 0.5$ and the mean and variance terms set by the averaging process.

The training network backward pass is an adaptation of backpropagation using the neuron and synapse equations above. To get the gradient at each synapse, we use the chain rule to compute

$$\frac{\partial E}{\partial \tilde{c}_{ij}} = \frac{\partial E}{\partial \tilde{n}_j}\frac{\partial \tilde{n}_j}{\partial \tilde{c}_{ij}}.$$

For the bias, a similar computation is made by replacing $\tilde{c}_{ij}$ in the above equation with $b_j$.

We can then differentiate Equation 3 to produce

$$\frac{\partial \tilde{n}_j}{\partial \tilde{c}_{ij}} = \frac{\tilde{x}_i s_{ij}}{\sigma_j \sqrt{2\pi}}e^{-\left(\frac{(\theta - \mu_j)^2}{2\sigma_j^2}\right)} - (\theta - \mu_j)\frac{\tilde{x}_i s_{ij}^2 - \tilde{x}_i^2 \tilde{c}_{ij}s_{ij}^2}{\sigma_j^3 \sqrt{2\pi}}e^{-\left(\frac{(\theta - \mu_j)^2}{2\sigma_j^2}\right)}. \tag{4}$$

As described below, we will assume that the synapse strengths to each neuron are balanced between positive and negative values and that each neuron receives 256 inputs, so we can expect $\mu$ to be close to zero, and $\mu$, $\tilde{n}_i$ and $\tilde{c}_{ij}$ to be much less than $\sigma$. Therefore, the right term of Equation 4 containing the denominator $\sigma_j^3$, can be expected to be much smaller than the left term containing the denominator $\sigma_j$. Under these conditions, for computational efficiency we can approximate Equation 4 by dropping the right term and factoring out the remainder as

$$\frac{\partial \tilde{n}_j}{\partial \tilde{c}_{ij}} \approx \frac{\partial \tilde{n}_j}{\partial \mu_j} \frac{\partial \mu_j}{\partial \tilde{c}_{ij}},$$

where

$$\frac{\partial \tilde{n}_j}{\partial \mu_j} = \frac{1}{\sigma_j \sqrt{2\pi}} e^{-\left(\frac{(\theta - \mu_j)^2}{2\sigma_j^2}\right)},$$

and

$$\frac{\partial \mu_j}{\partial \tilde{c}_{ij}} = \tilde{x}_i s_{ij}.$$

A similar treatment can be used to show that corresponding gradient with respect to the bias term equals one.

The network is updated using the loss gradient at each synapse and bias term. For each iteration, synaptic connection probability changes according to

$$\Delta \tilde{c}_{ij} = -\alpha \frac{\partial E}{\partial \tilde{c}_{ij}},$$

where $\alpha$ is the learning rate. Any synaptic connection probabilities that fall outside of the range $[0, 1]$ as a result of the update rule are "snapped" to the nearest valid value. Changes to the bias term are handled in a similar fashion, with values clipped to fall in the range $[-255, 255]$, the largest values supported using TrueNorth neuron parameters.

The training procedure described here is amenable to methods and heuristics applied in standard backpropagation. For the results shown below, we used mini batch size 100, momentum 0.9, dropout 0.5 [18], learning rate decay on a fixed schedule across training iterations starting at 0.1 and multiplying by 0.1 every 250 epochs, and transformations of the training data for each iteration with rotation up to $\pm 15°$, shift up to $\pm 5$ pixels and rescale up to $\pm 15\%$.

### 3.3 Mapping training network to deployment network

Training is performed offline, and the resulting network is mapped to the deployment network for hardware operation. For deployment, depending on system requirements, we can utilize an ensemble of one or more samplings of the training network to increase overall output performance. Unlike other ensemble methods, we train only once then sample the training network for each member. The system output for each class is determined by averaging across all neurons in all member networks assigned to the class. Synaptic connection states are set on or off according to $P(c_{ij} = 1) \equiv \tilde{c}_{ij}$, using independent random number draws for each synapse in each ensemble member. Data is converted into a spiking representation for input using $P(x_i = 1) \equiv \tilde{x}_i$, using independent random number draws for each input to each member of the ensemble.

### 3.4 Network initialization

The approach for network initialization described here allows us to optimize for efficient neuromorphic hardware that employs less than 2 bits per synapse. In our approach, each synaptic connection probability is initialized from a uniform random distribution over the range $[0, 1]$. To initialize synapse strength values, we begin from the principle that each core should maximize information transfer by maximizing information per neuron and minimizing redundancy between neurons. Such methods have been explored in detail in approaches such as *infomax* [19]. While the first of these goals is data dependent, we can pursue the second at initialization time by tuning the space of possible weights for a core, represented by the matrix of synapse strength values, $S$.

In our approach, we wish to minimize redundancy between neurons on a core by attempting to induce a product distribution on the outputs for every pair of neurons. To simplify the problem, we note that the summed weighted inputs to a pair of neurons is well-approximated by a bi-variate Gaussian distribution. Thus, forcing the covariance between the summed weighted inputs to zero guarantees that the inputs are independent. Furthermore, since functions of pairwise independent random variables remain pair-wise independent, the neuron outputs are guaranteed to be independent.

The summed weighted input to $j$-th neuron is given by Equation 1. It is desirable for the purposes of maintaining balance in neuron dynamics to configure its weights using a mix of positive and negative values that sum to zero. Thus for all $j$,

$$\sum_i s_{ij} = 0, \tag{5}$$

which implies that $E[I_j] \approx 0$ assuming inputs and synaptic connection states are both decorrelated and the bias term is near 0. This simplifies the covariance between the inputs to any two neurons on a core to

$$E[I_j I_r] = E\left[\sum_{i,q} x_i c_{ij} s_{ij} x_q c_{qr} s_{qr}\right].$$

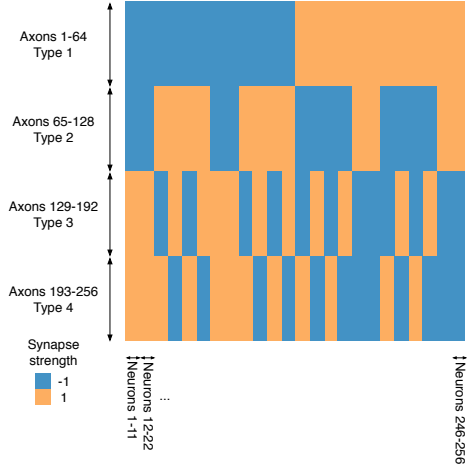

Figure 2: Synapse strength values depicted as axons (rows) × neurons (columns) array. The learning procedure fixes these values when the network is initialized and learns the probability that each synapse is in a transmitting state. The blocky appearance of the strength matrix is the result of the shared synaptic strength approach used by TrueNorth to reduce memory footprint.

Rearranging terms, we get

$$E[I_j I_r] = \sum_i c_{ij} s_{ij} c_{qr} s_{qr} E[x_i^2] + \sum_i c_{ij} s_{ij} \sum_{q \neq i} c_{qr} s_{qr} E[x_i x_q]. \tag{6}$$

Next, we note from the equation for covariance that $E[x_i x_q] = \sigma(x_i, x_q) + E[x_i]E[x_q]$. Under the assumption that inputs have equal mean and variance, then for any $i$, $E[x_i^2] = \rho$, where $\rho = \sigma(x_i, x_q) + E[x_i]E[x_q]$ is a constant. Further assuming that covariance between $x_i$ and $x_q$ where $i \neq q$ is the same for all inputs, then $E[x_i x_q] = \gamma$, where $\gamma = \sigma(x_i, x_q) + E[x_i]E[x_q]$ is a constant. Using this and equation (5), Equation 6 becomes

$$E[I_j I_r] = \rho \langle c_j s_j, c_r s_r \rangle + \gamma \sum_i c_{ij} s_{ij} (-c_{ir} s_{ir})$$
$$= \rho \langle c_j s_j, c_r s_r \rangle - \gamma \langle c_j s_j, c_r s_r \rangle$$
$$= (\rho - \gamma) \langle c_j s_j, c_r s_r \rangle.$$

So minimizing the absolute value of the inner product between columns of $W$ forces $I_j$ and $I_r$ to be maximally uncorrelated under the constraints.

Inspired by this observation, we apriori (i.e., without any knowledge of the input data) choose the strength values such that the absolute value of the inner product between columns of the effective weight matrix is minimized, and the sum of effective weights to each neuron is zero. Practically, this is achieved by assigning half of each neuron's s-values to $-1$ and the other half to 1, balancing the possible permutations of such assignments so they occur as equally as possible across neurons on a core, and evenly distributing the four possible axon types amongst the axons on a core. The resulting matrix of synaptic strength values can be seen in Figure 2. This configuration thus provides an optimal weight subspace, given the constraints, in which backpropagation can operate in a data-driven fashion to find desirable synaptic on/off states.

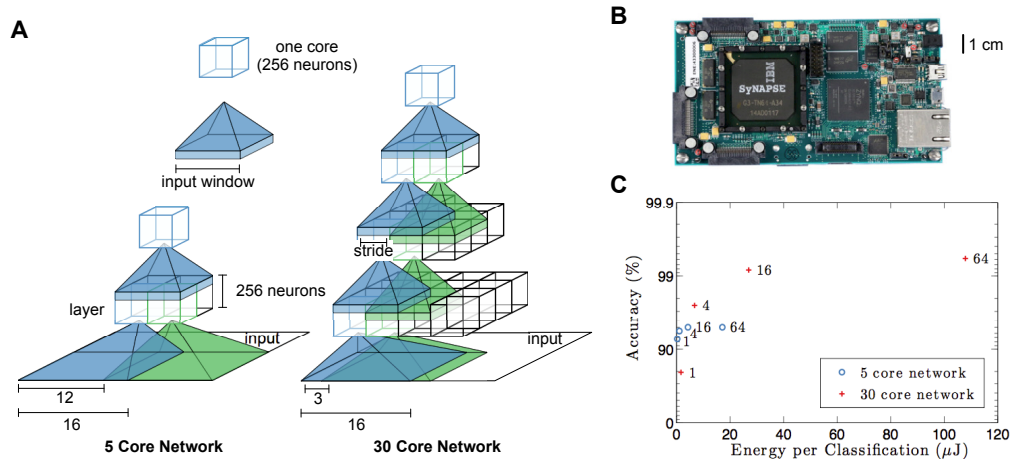

Figure 3:    A) Two network configurations used for the results described here, a 5 core network designed to minimize core count and a 30 core network designed to maximize accuracy. B) Board with a socketed TrueNorth chip used to run the deployment networks. The chip is $4.3$ cm$^2$, runs in real time (1 ms neuron updates), and consumes 63 mW running a benchmark network that uses all of its 1 million neuron [7]. C) Measured accuracy and measured energy for the two network configurations running on the chip. Ensemble size is shown to the right of each data point.

## 4   Network topology

The network topology is designed to support neurons with responses to local, regional or global features while respecting the "core-to-core" connectivity of the TrueNorth architecture – namely that all neurons on a core share access to the same set of inputs, and that the number of such inputs is limited. The network uses a multilayer feedforward scheme, where the first layer consists of input elements in a rows × columns × channels array, such as an image, and the remaining layers consist of TrueNorth cores. Connections between layers are made using a sliding window approach.

Input to each core in a layer is drawn from an $R \times R \times F$ *input window* (Figure 3A), where $R$ represents the row and column dimensions, and $F$ represents the feature dimension. For input from the first layer, rows and columns are in units of input elements and features are input channels, while for input from the remaining layers rows and columns are in units of cores and features are neurons. The first core in a given target layer locates its input window in the upper left corner of its source layer, and the next core in the target layer shifts its input window to the right by a *stride* of $S$. Successive cores slide the window over by $S$ until the edge of the source layer is reached, then the window is returned to the left, and shifted down by $S$ and the process is repeated. Features are sub-selected randomly, with the constraint that each neuron can only be selected by one target core. We allow input elements to be selected multiple times. This scheme is similar in some respects to that used by a convolution network, but we employ independent synapses for each location. The specific networks employed here, and associated parameters, are shown in Figure 3A.

## 5   Results

We applied the training method described above to the MNIST dataset [20], examining accuracy vs. energy tradeoffs using two networks running on the TrueNorth chip (Figure 3B). The first network is the smallest multilayer TrueNorth network possible for the number of pixels present in the dataset, consisting of 5 cores distributed in 2 layers, corresponding to 512 neurons. The second network was built with a primary goal of maximizing accuracy, and is composed of 30 cores distributed in 4 layers (Figure 3A), corresponding to 3840 neurons. Networks are configured with a first layer using $R = 16$ and $F = 1$ in both networks, and $S = 12$ in the 5 core network and $S = 4$ in the 30 core network, while all subsequent layers in both networks use $R = 2$, $F = 64$, and $S = 1$. These parameters result in a "pyramid" shape, where all cores from layer 2 to the final layer draw input

from 4 source cores and 64 neurons in each of those sources. Each core employs 64 neurons per core it targets, up to a maximum of 256 neurons.

We tested each network in an ensemble of 1, 4, 16, or 64 members running on a TrueNorth chip in real-time. Each image was encoded using a single time step (1 ms), with a different spike sampling used for each input line targeted by a pixel. The instrumentation available measures active power for the network in operation and leakage power for the entire chip, which consists of 4096 cores. We report energy numbers as active power plus the fraction of leakage power for the cores in use.

The highest overall performance we observed of $99.42\%$ was achieved with a 30 core trained network using a 64 member ensemble, for a total of 1920 cores, that was measured using 108 $\mu$J per classification. The lowest energy was achieved by the 5 core network operating in an ensemble of 1, that was measured using 0.268 $\mu$J per classification while achieving $92.70\%$ accuracy. Results are plotted showing accuracy vs. energy in Figure 3C. Both networks classified 1000 images per second.

## 6 Discussion

Our results show that backpropagation operating in a probabilistic domain can be used to train networks that naturally map to neuromorphic hardware with spiking neurons and extremely low-precision synapses. Our approach can be succinctly summarized as *constrain-then-train*, where we first constrain our network to provide a direct representation of our deployment system and then train within those constraints. This can be contrasted with a *train-then-constrain* approach, where a network agnostic to the final deployment system is first trained, and following training is constrained through normalization and discretization methods to provide a spiking representation or low precision weights. While requiring a customized training rule, the constrain-then-train approach offers the advantage that a decrease in training error has a direct correspondence to a decrease in error for the deployment network. Conversely, the train-then-constrain approach allows use of off the shelf training methods, but unconstrained training is not guaranteed to produce a reduction in error after hardware constraints are applied.

Looking forward, we see several avenues for expanding this approach to more complex datasets. First, deep convolution networks [20] have seen a great deal of success by using backpropagation to learn the weights of convolutional filters. The learning method introduced here is independent of the specific network structure beyond the given sparsity constraint, and could certainly be adapted for use in convolution networks. Second, biology provides a number of examples, such as the retina or cochlea, for mapping high-precision sensory data into a binary spiking representation. Drawing inspiration from such approaches may improve performance beyond the linear mapping scheme used in this work. Third, this approach may also be adaptable to other gradient based learning methods, or to methods with existing probabilistic components such as contrastive divergence [21]. Further, while we describe the use of this approach with TrueNorth to provide a concrete use case, we see no reason why this training approach cannot be used with other spiking neuromorphic hardware [4][5][6].

We believe this work is particularly timely, as in recent years backpropagation has achieved a high level of performance on a number tasks reflecting real world tasks, including object detection in complex scenes [1], pedestrian detection [2], and speech recognition [3]. A wide range of sensors are found in mobile devices ranging from phones to automobiles, and platforms like TrueNorth provide a low power substrate for processing that sensory data. By bridging backpropagation and energy efficient neuromorphic computing, we hope that the work here provides an important step towards building low-power, scalable brain-inspired systems with real world applicability.

**Acknowledgments**

This research was sponsored by the Defense Advanced Research Projects Agency under contracts No. HR0011- 09-C-0002 and No. FA9453-15-C-0055. The views, opinions, and/or findings contained in this paper are those of the authors and should not be interpreted as representing the official views or policies of the Department of Defense or the U.S. Government.

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
