[Reviews · NeurIPS 2015]

Submitted by Assigned_Reviewer_1

1) The authors should at least mention the algorithm in section 3.2 is deeply similar to the cited references

[14] J. Zhao, J. Shawe-Taylor, and M. van Daalen, "Learning in stochastic bit stream neural networks," Neural Networks, vol. 9, no. 6, pp. 991 - 998, 1996. and [15] Soudry, Daniel, Itay Hubara, and Ron Meir. "Expectation backpropagation: Parameter-free training of multilayer neural networks with continuous or discrete weights." Advances in Neural Information Processing Systems. 2014.

in its final form (which is a hybrid of both algorithms), and was derived using identical approximations.

2) The first paragraph in the discussion gives the impression the "constrain-then-train" methodology (learning a weight distribution and then sampling from it to produce a fixed network ensemble at test time) is novel, even though it was already suggested and tested in ref [15].

3) Novelty claim (i) in the last paragraph of the introduction is also misleading. Ref. [15] "expectation backpropgation" algorithm is a backpropgation update rule (using expectation propagation), implemented for binary neurons (i.e., with a sign activation function, in which zero input is never reached exactly) with binary weights, and can easily have constrained connectivity.

4) The cited previous state-of-the-art are wrong. To the best of my knowledge, the previous best results for binary weights (and neurons) were achieved by the algorithm in ref. [15], which was tested on the mnist (http://arxiv.org/pdf/1503.03562v3.pdf) without data augmentation, as was done in this paper.

%%% Edited after author's feedback Thank you for agreeing to address these concerns.
Summary: This paper has quite strong and novel experimental results, in terms of accuracy and power consumption. However, both the general methodology and the algorithm itself are not as novel as claimed in the paper.

Submitted by Assigned_Reviewer_2

The authors propose a way to reconcile the need for high precision of continuous variables in backpropagation algorithms with the desire to use spike-based neural network devices with binary synapses. They introduce the idea of considering the analog variables as probabilities that are sampled to map the neural network used off-line for training to the spiking network chip used for deployment of the application. They show how to adapt the backpropagation algorithm to be compatible with this strategy and validate the method proposed on the MNIST benchmark. They map the trained network onto two different architectures, optimizing for size and accuracy, and present performance figures measured from the chip. The quality of the manuscript is very high, as well as its clarity. The work is quite original and can be useful for other approaches and other hardware platforms. BTW, in reviewing the different HW platforms available, the authors should mention also Qiao et al., Frontiers in Neuroscience 2015, and the paper at http://arxiv.org/abs/1506.05427 The significance of the work is not highlighted in the paper. It is not clear how the method proposed will be useful to deploy the TrueNorth chip in practical applications, and in what specific applications. It is not clear if and to what extent this specific method has advantages over other alternative options, such as those cited in [9] to [12].

Summary: The authors present a method for applying backpropagation to spiking neural networks with binary synapses and validate it by successfully mapping the trained network onto the TrueNorth device. The paper is well structured, clear and to the point. The results are convincing, but lack some details (e.g. how are the inputs converted into spikes, what are the frequencies used, the shared weight values, the neuron models, etc.)

Submitted by Assigned_Reviewer_3

The authors present an interesting paper about a backpropagation training method using spike probabilities and takes into account the hardware constraints of a particular platform which has spiking neurons and discrete synapses. This is a topic of current interest in mapping deep networks to multi-neuron hardware platforms.

Quality: The proposed training method is useful especially in consideration of new multi-neuron hardware platforms with constraints.

Clarity: The paper is easy to read. The claims made at the end of Section 1 can be reworded because as it stands, if not read carefully, suggests that the paper proposes for the first time a training methodology that employs spiking neurons, synapses with reduced precision. I would not necessarily put the demonstration of running this network on a specific hardware platform, in this case TN, as a novel feature to be included in this case. Running the network on TN is a validation of the training method.

In Section 2, what does it mean for 0.15 bits per synapse? The network topology is not clearly described in one place. How much longer is the training time because of the probabilistic synaptic connection update? Why is the result from a single ensemble of the 30 core network so low especially when compared to the 5 core network? Why are the results of the different ensembles of the 5 core network about the same? What are the spike rates for the inputs during testing? Is the input a spike train? Ref 12 on line 399 includes constraints (e.g. bias=0) during training so it is not just a train-then-constrain alone approach (typo on this line, "approach approach").

Originality:

Can the authors provide a discussion or comparison with other spiking backprop-type rules like SpikeProp?

Significance: The development of new training methods that consider constraints of hardware platforms is of great interest, The constraints which are considered during training in this paper are based on the TN architecture, these constraints might not all apply to other platforms. If the results are based on a 64 ensemble, that means more hardware resources have to be dedicated to obtain the accuracy needed of the network.
Summary: An interesting paper about a training method using backpropagation which takes into account the hardware constraints of a particular platform which has spiking neurons and discrete synapses.

Author Feedback
Author rebuttal: We thank the reviewers for their insightful suggestions and will work to address all concerns for the final manuscript. There is unfortunately not space here to individually address each comment, but we provide select responses here.

R1. "The significance of the work is not highlighted in the paper. It is not clear how the method proposed will be useful to deploy the TrueNorth chip in practical applications, and in what specific applications."

We believe this is a foundational paper linking the backpropagation approach with existing neuromorphic hardware. Our current focus is to build feedforward classifier networks, which is an important building block for many real time applications that we plan to deploy on TrueNorth.

R1. "It is not clear if and to what extent this specific method has advantages over other alternative options, such as those cited in [9] to [12]."

We will emphasize the advantages of this approach over previous work, which is to train networks for neuromorphic hardware deployment using low precision synapses, spiking neurons and sparse connectivity all in a single learning approach.

R1. BTW, in reviewing the different HW platforms available, the authors should mention also Qiao et al., Frontiers in Neuroscience 2015, and the paper at http://arxiv.org/abs/1506.05427

We will cite these papers in our revised manuscript.

R1. "The results are convincing, but lack some details (e.g. how are the inputs converted into spikes, what are the frequencies used, the shared weight values, the neuron models, etc.) "
AND
R2. "What are the spike rates for the inputs during testing? Is the input a spike train?"
AND
R5. "The description of the network operation should be more clear; for example, how does the network run 1,000 images per second if the chip has 1ms neuron updates (line 343)?"

We will increase the clarity of these sections by describing the spike creation process, neuron and network operation in more detail.

R2. "The claims made at the end of Section 1 can be reworded because as it stands, if not read carefully, suggests that the paper proposes for the first time a training methodology that employs spiking neurons, synapses with reduced precision."

We will reword and clarify this section to emphasize that we are doing offline training to create networks to deploy in systems with spiking neurons and synapses with reduced precision. As far as we are aware, this is the first work to directly use backprop to train networks that are deployed with all three constraints: Binary neurons, extremely low precision synapses and limited neuron fan-in.

R3. "Ref 12 on line 399 includes constraints (e.g. bias=0) during training so it is not just a train-then-constrain alone approach (typo on this line, "approach approach")."

We will correct the typo and properly describe Ref 12.

R3. "Additionally, it would be helpful to compare the total training time varying the number of ensemble members in order to achieve the same level of accuracy."

We use ensembles a bit differently from other approaches and we will briefly explain how in the text. Rather than training multiple networks, we train once and take advantage of the probabilistic synapse representation to draw multiple deployment networks using stochastic sampling from the set of trained synapse probabilities. Thus, it takes the same time to train a network with 1 ensemble members as 64 ensemble members.

R6. "The authors should at least mention the algorithm in section 3.2 is deeply similar to [14, 15] in its final form (which is a hybrid of both algorithms), and was derived using identical approximations."

We will clarify similarities between this manuscript and previous works.

R6. "Lines 395-406 give the impression the "constrain-then-train" methodology (learning a weight distribution and then sampling from it to produce a fixed network ensemble at test time) is novel even though it was already suggested and tested in ref [15]."

We will adjust the wording here to clarify.

R6. "Novelty claim (i) on Lines 69-71 is also misleading. Ref. [15] "expectation backpropgation" algorithm is a backpropgation update rule (using expectation propagation), implemented for binary neurons (i.e., with a sign activation function, in which zero input is never reached exactly) with binary weights, and can have constrained connectivity."

While constrained connectivity is not demonstrated in Ref 15, there is no reason the method described there-in couldn't be used in such a context. We will note this in our manuscript.

R6. "The cited previous state-of-the-art on lines 398-400 are wrong. To the best of my knowledge, the previous best results for binary weights (and neurons) were achieved by the algorithm in ref. [15], which was tested on mnist (http://arxiv.org/pdf/1503.03562v3.pdf)."

We will correct our manuscript to properly reference these results.